# On the Use of Polymer Inclusion Membranes for the Selective Separation of Pb(II), Cd(II), and Zn(II) from Seawater

**DOI:** 10.3390/membranes13050512

**Published:** 2023-05-12

**Authors:** Mariana Macías, Eduardo Rodríguez de San Miguel

**Affiliations:** Departamento de Química Analítica, Facultad de Química, Universidad Nacional Autónoma de México (UNAM), Ciudad Universitaria, Mexico City 04510, Mexico; mariana.19.l@hotmail.com

**Keywords:** polymer inclusion membranes, selective separation, cadmium, lead, zinc, seawater

## Abstract

The synthesis and optimization of polymeric inclusion membranes (PIMs) for the transport of Cd(II) and Pb(II) and their separation from Zn(II) in aqueous saline media are presented. The effects of NaCl concentrations, pH, matrix nature, and metal ion concentrations in the feed phase are additionally analyzed. Experimental design strategies were used for the optimization of PIM composition and evaluating competitive transport. Synthetic seawater with 35% salinity, commercial seawater collected from the Gulf of California (Panakos^®^), and seawater collected from the beach of Tecolutla, Veracruz, Mexico, were employed. The results show an excellent separation behavior in a three-compartment setup using two different PIMs (Aliquat 336 and D2EHPA as carriers, respectively), with the feed phase placed in the central compartment and two different stripping phases placed on both sides: one solution with 0.1 mol/dm^3^ HCl + 0.1 mol/dm^3^ NaCl and the other with 0.1 mol/dm^3^ HNO_3_. The selective separation of Pb(II), Cd(II), and Zn(II) from seawater shows separation factors whose values depend on the composition of the seawater media (metal ion concentrations and matrix composition). The PIM system allows S(Cd) and S(Pb)~1000 and 10 < S(Zn) < 1000, depending on the nature of the sample. However, values as high as 10,000 were observed in some experiments, allowing an adequate separation of the metal ions. Analyses of the separation factors in the different compartments in terms of the pertraction mechanism of the metal ions, PIMs stabilities, and preconcentration characteristics of the system are performed as well. A satisfactory preconcentration of the metal ions was observed after each recycling cycle.

## 1. Introduction

The management of marine systems has become a global challenge due to its importance and impact on the economy and social well-being. Preserving the oceans and marine resources is determinant both for the health of the planet and for the economy and food security of millions of people worldwide [1,2]. Regrettably, the pollution of the aquatic environment impacts the survival of organisms by altering their physiology, e.g., organs and their stages of development. Many of these effects may be present over several generations and may have large effects on the population and ecosystem levels, even affecting biological diversity [3].

The presence of heavy metals in the environment has grown due to their extensive use in some industrial and agricultural activities. The term heavy metal refers to any metallic chemical element that has a high relative density compared to water and is potentially toxic to animals, plants, or humans, even at very low concentrations [4]. Although these metals are terrestrial products, they end up in the sea through effluents and sewage or are discharged directly into seawater [5,6]. Their concentrations vary widely according to different seawater latitudes and depths and can be strongly influenced by freshwater discharges from highly polluted rivers [3].

In particular, cadmium and lead are heavy metals that represent a high toxicological risk due to the detrimental effects that their presence has on human health and marine biota. Cadmium is found in the earth’s crust in small amounts and can be extracted through the production of copper, lead, and zinc. It can also be emitted into the environment by burning fossil fuels such as coal and oil or by burning waste. It has been used in the production of batteries, and in enamels and metal coatings [7]. Cadmium concentrations in seawater range from 0.001 to 0.1 μg/dm^3^ in open oceans and from 0.008 to 0.25 μg/dm^3^ in coastal waters and semi-enclosed seas [8,9]. Its maximum permissible concentration in salt water indicated by the EPA is 2.9 × 10^−7^ mol/dm^3^ (33 μg/dm^3^) [10]. On the other hand, lead is found in the earth’s crust in small quantities; environmental exposure comes mainly from the old use of gasoline and paints with its high content, production of ceramics, batteries, welding, pipes, ammunition, aircraft with spark ignition, or through its own mining production. Lead concentrations in seawater range from 0.002 to 0.36 μg/dm^3^ in open oceans [11] and from 0.5 to 1 μg/dm^3^ in coastal waters and semi-enclosed seas [9]. Its maximum permissible concentration in salt water indicated by the EPA is 10.14 × 10^−7^ mol/dm^3^ (210 μg/dm^3^) [10]. As for zinc, it is generally considered that Zn^2+^ is the species primarily responsible for eliciting a toxic response in aquatic organisms. Typically, inorganic and organic complexes ameliorate the uptake and toxicity of zinc by reducing the concentration of Zn^2+^. An exponential, inverse relationship exists between water hardness and the uptake and toxicity of zinc. An algorithm describing this relationship has been used to calculate a hardness-modified zinc guideline value for protecting aquatic ecosystems in North America [12]. Zinc concentrations in estuaries and shorelines are often greater than 0.5 μg/dm^3^ and up to 379 μg/dm^3^ [13].

Although different technologies for removing heavy metal ions from waters are currently available, e.g., chemical precipitation, ion-exchange, adsorption, coagulation–flocculation, flotation, photocatalysis, electrochemical methods [14,15], and polymer-modified magnetic nanoparticles [16], membrane technology is becoming more advanced due to environmental friendliness, economics, and ease of use [17]. One of the main advantages over other separation technologies is its versatility, as membranes can be integrated with other processes, including different membrane techniques, without much difficulty [18,19]. It can separate at the molecular scale, which means that a large number of separation needs can be met by membrane processes, and a wide range of polymers and inorganic media can be used as membranes, providing a great deal of control over separation selectivities [20,21,22]. Moreover, different kinds of modules can be selected to fit the desired application [19,21]. Membrane processes are also energy-efficient and potentially better for the environment since they require the use of relatively simple and non-harmful materials and generate nontoxic reaction by-products [23]. Furthermore, they can recover minor but valuable components from a mainstream without substantial energy costs.

Polymer inclusion membranes (PIMs) are a highly promising technique for metal separation and recovery due to their outstanding performance compared to other types of liquid membranes. They have several advantages over other separation processes, including high stability, selectivity, efficiency, and durability [24]. PIMs are also flexible, low cost, and easily satisfy environmental pollution regulations. Additionally, they offer controlled membrane permeability, which is important for selective separation and recovery of metals [25]. PIMs have higher performance than expected, and when production conditions are examined, they are more advantageous than other processes [26]. Furthermore, the reusability of the membranes and their large surface area/volume ratio is an important feature related to their industrial uses [26]. However, PIMs also have some disadvantages, such as low flux and fouling, whereby further research is required to optimize their performance and overcome their limitations.

PIMs have effectively been employed in separation science, e.g., for the hydrometallurgical separation of Co(II) from Mn(II) [27], the transport of Zn(II), Fe(II), and Fe(III) ions from chloride aqueous solutions [28], the separation of Ag(I) from Cu(II) [29], the separation of In(III) from Cu(II) in hydrochloric acid medium [30], the selective separation of scandium (Sc) from other rare earth metals (Y, La, Nd, Eu, Dy) [25], the transport of Pd(II) over Pt(IV), Mn(II), Ni(II), and Fe(III) [31], the separation of Pd(II) and Rh(III) from chloride solutions [32], the possibility of cadmium(II) and lead(II) ion separation [33], and the selective removal of cobalt(II) from aqueous chloride solutions containing nickel(II) and lithium(I) [34], among many other applications.

The results of research conducted in recent years clearly show that PIMs can positively be used to remove hazardous metal ions from various types of wastewaters and to recover noble metal ions from waste, e.g., from waste electrical and electronic equipment. However, the successful utilization of certain PIMs in laboratory conditions does not guarantee their effectiveness and resistance to long-term use on an industrial scale [35]. The implementation of PIMs in industry remains challenging, and only with an understanding of the membrane process on a fundamental level can advanced separation technologies be implemented to ensure the sustainability of metal resources [36]. In particular, it has been observed that in complex mixtures (Zn(II), Cd(II), Co(II), Cu(II), and Ni(II)) the values of the initial fluxes depend on the composition of the multicomponent solution and that the selectivity coefficients decrease with the increasing amount of ions in the mixture as well as the recovery of the different metals [37].

As for sea water, Paredes et al. [38] previously reported the use of a PIM composed of cellulose triacetate (CTA) and two carriers (LIX 54-100 and Cyanex 923) for the extraction of lithium. Djunaidi et al. [39], and Djunaidi and Wahyuni [40] have evaluated the transport of sodium through a PIM containing Aliquat 336 (methyltrioctylammonium chloride)-TBP (tributylphosphate) mixtures as carrier or dibutyl ether, Aliquat 336, D2EHPA (di-(2-ethylhexyl)-phosphate), thenoyl trifluoroacetone (HTTA), TBP, and eugenol (PE), respectively, for the desalination of seawater. Khaldoun et al. [41] proposed a PIM-based device with the ionic liquid trihexyl (tetradecyl) phosphonium chloride (THTDPCl) for Cd monitoring in seawater. This study showed that the efficiency of the PIM system is not affected either by high salinity or the presence of large amounts of other ions and can thus facilitate Cd monitoring in seawater samples. López-Guerrero et al. [42] developed a PIM for the simultaneous determination of Cu(II), Ni(II), and Cd(II) ions from natural waters of different salinities using pyridine-2-acetaldehyde benzoylhydrazone (2-APBH) as extractant.

Nonetheless, as most of the works on separations with PIMs are focused on synthetic model solutions [35] using mainly fixed quantities of the metal ions, e.g., equimolar, in one single separation step, i.e., one feed and strip phase, there is a need for the development of new schemes to solve more complex separation problems. In this regard, innovative sequential transport schemes in which the stripping solution is replaced to attain the desired recoveries and selectivities have been reported for the separation of Pt(IV), Pd(II), and Rh(III) [36], and a two-stage process with different carriers in each step has also been used for the separation of Zn(II) and Ag(I) both in model solutions and in a solution after leaching silver-oxide waste batteries [43]. Three compartments’ schemes have also been implemented with two sequential stripping solutions for As(V) separation from Cu(II) [44] and for the simultaneously recovery and separation of Cu(II) from two different feed solutions (copper–nickel–cobalt and copper–zinc) using only one stripping phase [45].

Recent reports concerning the pollution status of heavy metals in surface sea water and sediments of the Tianjin Coastal Area (North China) point out potential ecological risk of Cd [46], and although the concentration of Pb did not exceed the Seawater Quality Standard in the Jieshi Bay (Shanwei, China), attention has been addressed to decision-makers to be alert to the Pb content in fish because of bioaccumulation from seafood product consumption [47]. To the best of our knowledge, the selective transport of heavy metals from seawater using integral and innovative PIM separation schemes has not been evaluated so far. For this reason, in the present work the optimization of PIMs for the individual transport of Cd(II) and Pb(II) from added synthetic saline media and two real samples of commercial seawater and seawater from Tecolutla beach (Veracruz, México) simulating polluted media with a serious potential ecological risk index [46,47] is described. A simultaneous separation system of Cd(II), Pb(II), and Zn(II) in a three-compartment process employing one feed phase and two stripping phases with two different PIMs is additionally presented to show the potential use of novel separation setups in the field of remediation of marine systems. The impact of the nature of the matrix (synthetic and real) and the concentration of the metal ions on the separation factors is examined as well.

## 2. Materials and Methods

### 2.1. Reactives and Equipment

Aqueous solutions of the metal cations Cd(II), Pb(II), and Zn(II) were prepared from a standard solution for ASS (1000 mg/L, Sigma Aldrich Corporation, St. Louis, MO, USA) using distilled and deionized water. Cellulose triacetate (CTA 100%, Sigma Aldrich), tris(2-ethylhexyl) phosphate (TEHP 97%, Sigma Aldrich), tris(2-butoxyethyl) phosphate (TBEP, 94%, Sigma Aldrich), 2-nitrophenyl octyl ether (2NPOE 99%, Sigma Aldrich), di2(ethyl-hexyl) acid phosphoric acid (D2EHPA 97%, Rhodia, La Défense, France), tricaprylylmethylammonium chloride and trioctylmethylammonium chloride mixture (Aliquat 336 > 97%, Sigma Aldrich), absolute ethyl alcohol (99.9%, J.T. Baker, Phillipsburg, NJ, USA), and dichloromethane (99% J.T. Baker) were used to prepare the membranes. All reagents were employed as received. Hydrochloric acid (HCl 37%, Sigma Aldrich) and nitric acid (HNO_3_ 65%, Merck, Rahway, NJ, USA) solutions were used as stripping phases. 2-amino-2-hydroxymethyl-1.3-propanediol (>99%, Goldbio, St Louis, MO, USA), was used in the preparation of buffer solutions. Sodium chloride (NaCl ≥ 99%, J.T. Baker), sodium fluoride (NaF ≥ 99%, Sigma Aldrich), potassium chloride (KCl ≥ 99%, Sigma Aldrich), sodium sulfate (Na_2_SO_4_ ≥ 99%, Sigma Aldrich), magnesium chloride (MgCl_2_ ≥ 98%, Sigma Aldrich), anhydrous calcium chloride (CaCl_2_ ≥ 96%, Sigma Aldrich), boric acid (H_3_BO_3_ ≥ 99.97%, Sigma Aldrich), strontium chloride hexahydrate (SrCl_2_ 99%, Sigma Aldrich), anhydrous sodium bicarbonate (Na_2_CO_3_ ≥ 99.7%, Sigma Aldrich), and potassium bromide (NaBr ≥ 99%, Sigma Aldrich) were employed for the preparation of synthetic seawater (35% salinity). Deep seawater (at least 20 m deep) commercially collected from the Gulf of California, subjected to double cold sterilizing microfiltration at 0.22 microns and pH = 8.2 [48], and seawater collected from the beach of Tecolutla (Veracruz, México) with pH = 8.1 and conductivity of 51.65 mS/cm (T = 23.7 °C) were employed. For this, surface samples (<1 m) were collected using pyrex bottles with air-tight seals leaving a headspace of about 1% of the total volume following the procedure reported by Yeats [49]. The samples were preserved in refrigeration until their use as feed phases, minimizing the storage time; i.e., they were used within the next 5 days.

The differential pulse voltammetry (DPV) technique with a Sensit Smart portable potentiostat with the autonomous EmStat Pico module was employed for metal ion determinations as included in the device (PalmSens Compact Electrochemical Interfaces, Utrecht, The Nederland). A Metromhm 620 pH-meter with a Cole-Parmer glass combination electrode was used for all pH measurements. The 797 VA computrace computer software (Metrohm Ion analysis, version 1.2, Herisau, Switzerland) and PStouch software (PalmSens Compact Electrochemical Interfaces, Utrecht, The Nederland) were used for measurement, recording, and processing of electrochemical data.

A Perkin Elmer (Waltham, MA, USA) Spectrum GX FTIR spectrometer coupled with a diamond ATR sampling accessory (DuraSampl IR II from SensIR Technologies, Danbury, CT, USA) was used for acquisition of FTIR spectra using the manufacturer’s software (AutoImage v. 5.0). Infrared maps were obtained by reflection infrared mapping microscopy (RIMM) using a Perkin–Elmer GX-FTIR spectrometer coupled to an Autoimage FTIR microscope with Autoimage v. 5.0 software. Maps were performed using 100 μm × 100 μm aperture in 2000 μm × 2000 μm areas, 4 cm^−1^ resolution, 30 scans per point, in the 4000–700 cm^−1^ region, and all measures were zero-corrected with the 3840 cm^−1^ line as reference.

### 2.2. PIM Preparation

PIMs were prepared by the previously reported casting-evaporation procedure [50]; i.e., specific amounts of plasticizer, extracting agent and polymeric base were weighed and dissolved in 10 cm^3^ of dichloromethane. In the case where the extracting agent was D2EHPA, 1 cm^3^ of ethyl alcohol was added to improve its solubility [51]. The mixture was stirred for at least 2 h and at the end of this lapse transferred to a 5 cm in diameter Petri Pyrex^®^ (Corning, New York, NY, USA) glass dish and left to stand for 48 h at room temperature until complete evaporation of the solvent. The homogeneous and transparent film formed was soaked with distilled water to facilitate its handling and to be able to be mounted in the transport cells. The specific compositions of the employed membranes are reported throughout the text.

### 2.3. Transport Experiments

PIMs were placed as an intermediate barrier in a home-made PVC permeation cell. The exposed area of the membrane was 4.9 cm^2^; each of the compartments had a capacity of 100 cm^3^ in which the corresponding feed and stripping solutions were placed and stirred at 450 rpm by mechanical motors. The cell arrangement corresponding to a two-compartment setup (Figure 1A) was used during the optimization experiments for the quantitative transport of each heavy metal (feed phase, stripping phase). The selective separation experiments were performed in a three-compartment permeation cell according to the arrangement shown in Figure 1B. The selectivity studies were carried out using synthetic seawater, whose composition is reported in ASTM D1141—98 [52] and Millero [53]. Experiments were performed in duplicate. The average RSD was within 5%.

### 2.4. PIM Optimization

The PIM optimization process for the transport of Cd(II) was achieved using a Box–Behnken design matrix in which multiple membrane compositions were tested, using 0.1 mol/dm^3^ NaCl pH = 6.5 aqueous solution as feed phase and 0.1 mol/dm^3^ HNO_3_ as stripping phase. The selection of the membrane components for this case was made considering the information reported by Briones Guerash Silva [54]. Table 1 shows the coding of the variables for this design.

As for the case of Pb(II), the composition of the membrane used was 0.05 g CTA, 0.0302 g plasticizing agent, and 0.0503 g D2EHPA, using a 0.1 mol/dm^3^ NaCl, pH = 6.5 aqueous solution as feed phase and 0.1 mol/dm^3^ HNO_3_ as stripping phase as reported by Salazar-Alvarez G. et al. [55]. In this case, three possible plasticizers, TBEP, NPOE, and TEHP, were tested. Once the membranes were optimized, subsequent experiments were performed using such membranes.

### 2.5. Effect of the Increase in NaCl Concentration in the Feed Phase

As for these experiments, the optimized membranes were used to evaluate the effect of NaCl concentration in the feed phase at 0.1, 0.3, and 0.5 mol/dm^3^ to arrive at the approximate concentration of NaCl present in seawater. The feed phase was 1 × 10^−4^ mol/dm^3^ metal(II), pH = 6.5, with variable NaCl content, and 0.1 mol/dm^3^ HNO_3_ was used as stripping phase.

### 2.6. Effect of pH in the Feed Phase

The effect of changing the pH in the feed phase from 6.3 to 8.2 was evaluated by employing 0.01 mol/dm^3^ TRIS-HCl buffer to cover the pH range in which ocean water bodies can be found, using 1 × 10^−4^ mol/dm^3^ metal(II), pH = 6.5, and 0.5 mol/dm^3^ NaCl as feed phase and 0.1 mol/dm^3^ HNO_3_ as stripping phase.

### 2.7. Effect of Matrix Nature

The effect of changing the feed phase matrix was evaluated using 35% salinity synthetic seawater prepared as reported by Millero [53]. Panakos commercial seawater (pH = 8.2) and seawater from the beach of Tecolutla (pH = 8.1) were employed as well. The experiments were carried out using 1 × 10^−4^ mol/dm^3^ metal (II) in seawater as feed phase and 0.1 mol/dm^3^ HNO_3_ as stripping phase.

### 2.8. Selective Separation of Cd(II), Pb(II), and Zn(II)

Experiments were performed using at the first feed-stripping interface (S1) a 0.05 g CTA, 0.06 g Aliquat 336, and 0.0325 g NPOE PIM composition, and at the second feed-stripping interface (S2) a 0.05 g CTA, 0.0453 g TEHP, and 0.0503 g D2EPHA PIM composition. Synthetic seawater, commercial seawater (Panakos), and seawater from the beach of Tecolutla Ver. Mex were employed as described in each experiment. A feed phase (F) with variable concentrations of Zn(II), Pb(II), and Cd(II) was employed. Using the three-compartment array previously described (Figure 1B), F was placed at the central compartment and two different stripping phases (S1 and S2) were placed on both sides: one solution with 0.1 M HCl + 0.1 mol/dm^3^ NaCl, and the other with 0.1 mol/dm^3^ HNO_3_, respectively (Figure 1).

#### 2.8.1. Effect of the Initial Concentrations of the Metal Ions

The effect of concentration on system performance was evaluated through a full 2^3^ factorial DOE. The levels and factors used for the experiments are shown in Table 2.

#### 2.8.2. PIM Stability and Preconcentration

The stability of the membranes and the preconcentration capacities of the systems were evaluated through successive experiments in which the feed phase was renewed without changing the stripping phases. The experiments were carried out over 3 cycles using commercial seawater and seawater from Tecolutla Ver. Mex.

## 3. Results and Discussion

### 3.1. PIM Optimization

Figure 2 presents the transport profiles obtained for each of the DOE experiments carried out for the optimization of Cd(II) transport. They graphically represent the fraction of the metal species present in the feed phase ($\Phi_{F}$, depletion) or the stripping phase ($\Phi_{S}$, recovery) at different pertraction times. The fraction of species is defined as
(1)ΦF=CFiCF°ΦS=CSiCF°
where $C_{F}^{i}$ stands for the concentration of the metal that remains in the feed phase, $C_{S}^{i}$ for the concentration of metal in the receiving phase, both after an extraction time t, and $C_{F}^{{}^{\circ}}$ for the initial concentration of metal in the feed phase.

Table 3 shows a summary of the results obtained including the composition of the membranes in coded values and their respective final species fractions for each phase. According to these results, membranes 8 and 12 showed the highest extraction capabilities percentages. However, to be able to obtain a complete description of the performance of the system as a function of PIM composition, an exponential function was used to adjust the profiles of the fraction of species of the feed and stripping phases (Figure 2), according to the following:(2)$$\Phi =Ae^{-t/d}+y_{0}$$
where *t* is the transport time, the parameter *A* defines the intersection of each curve with the ordinate, the parameter *d* provides information about the variation of the slope, which is indicative of the speed of transport, and *y*_0_ provides the limit value at which the profiles tend to the ordinate axis, i.e., the maximum or minimum concentration that is reached in the feed and stripping phases, respectively. Finally, to have the total description considering both phases, a desirability function, *D*, was obtained by the combination of the following functions as previously reported [56]:(3)$$G_{feed=\frac{1}{y_{0}d}}$$
(4)$$G_{stripping=\frac{y_{0}}{d}}$$
(5)$$D={\left\{G_{feed,0}^{I_{1}}G_{stripping,0}^{I_{2}}\right\}}^{1/\sum_{1}^{2}I_{k}}$$
where *G_feed_,*_0_ and *G_stripping_*,_0_ are normalized functions and *I_k_* are impact coefficients ranging between 1 and 5. In the present case, the value of 3 of all *I_k_* was assumed as the goal was to transport Cd(II) from feed to strip solution with minimum accumulation inside the membrane.

In Figure 3, contour plots of the desirability function are shown. Intermediate to high NPOE content with high Aliquat 336 and low CTA content makes the best PIMs. According to the response surface, an optimal desirability value of 0.81 may be achieved at −0.60 CTA, 1.0 NPOE, and 1.0 Aliquat 336 PIM coded composition. However, this PIM was fragile and difficult to manipulate. Membrane 8 of Table 3 (−1.0 CTA, 0.0 NPOE, 1.0 Aliquat 336) reached a desirability of 0.77, representing a non-significant difference with respect to the optimum value. As it has been reported that an excess of plasticizing agent can create an additional barrier for the transport of the metal ion, decreasing the mechanical resistance of the membrane [57] and increasing the possibility that it exudes towards some of the aqueous phases of the system, thus decreasing stability, adequate manipulation characteristics of the PIM at this composition were attained. Further experiments were performed using the composition levels of membrane 8.

In the case of the selection of the appropriate plasticizer for the transport of lead, even though the three compounds used showed a high and similar extraction efficiency (Figure 4), TEHP was selected because it allows obtaining the highest extraction percentages even when increasing the amount of NaCl in the medium (see Section 3.3).

### 3.2. Transport Mechanisms

The transport of Cd(II) through PIMs in the studied chloride media using Aliquat 336 (${CH}_{3}{\left(C_{8}H_{17}\right)}_{3}N^{+}{Cl}^{-}$) can be described according to the following reaction [58]:
(6)$${CdCl}_{3}^{-}+\overset{-}{{CH}_{3}{\left(C_{8}H_{17}\right)}_{3}N^{+}{Cl}^{-}}↔\overset{-}{{CH}_{3}{\left(C_{8}H_{17}\right)}_{3}N^{+}{CdCl}_{3}^{-}}+{Cl}^{-}$$from which the driving force of transport is the gradient in chloride ions between the feed and stripping phases. The anion exchange mechanism then allows the migration of a negative charge complex of the metal ion from the feed to the stripping phase with the simultaneous migration of chloride anions in reverse direction, i.e., the feed phase is depleted from the metal while it is enriched in chloride ions. However, to be precise, historically the term anion exchange originates from salt metathesis reactions, in which the anion initially present in the organic phase is exchanged by another anion initially present in the aqueous phase. The extraction of metals by basic extractants is usually assumed to be facilitated by the formation of the anionic ${MX}_{y}^{n-y}$ complex (*M*: metal, *X*: complexing agent, e.g., chloride) in the aqueous phase. However, experimental evidence has shown that the presence of the negatively charged anions in the aqueous phase is not mandatory, as the metal can be extracted to the organic phase despite that such species are not present in the aqueous phase. A new extraction model has then been provided relying on the hypothesis that the metal species least stabilized in the aqueous phase by hydration (i.e., the metal species with the lowest charge density) is extracted more efficiently than the more water- stabilized species (i.e., species with higher charge densities). Once it is transferred to the organic phase, the extracted species can undergo further Lewis acid−base adduct formation reactions with the chloride anions available in the organic phase to form negatively charged chloro-complexes in that phase; i.e., the anionic compounds are directly formed in the organic phase without requiring to be present initially in the aqueous phase [59]. Despite this and for the sake of simplicity, further explanations will be given considering exclusively Equation (6).

As for Pb(II) transport, the following transport reaction in PIMs with D2EHPA (*RH*) has been reported [60]:(7)$${Pb}^{2+}+2\overset{-}{{\left(RH\right)}_{2}}↔\overset{-}{PbR_{2}.2(RH)}+2H^{+}$$

However, as chloride is present in the feed phase, lead chloro-complexes ${PbCl}_{n}^{2-n}$ are formed, and competition among the free ion and those complexes is established. The results here obtained clearly show that this inhibitory effect is compensated by the effectiveness in the extraction reaction. Additionally, the counter-transport of hydronium ions is the driving force of the process, i.e., the feed phase is depleted from the metal while an increase in hydronium concentration is established.

### 3.3. Effect of NaCl in the Feed Phase

Figure 5 shows that there is no significant change (greater than 5%) in the transport profiles and the extraction efficiency of Cd(II) when increasing the concentration of NaCl in the feed phase. The same behavior was obtained by López-Guerrero [42]. This behavior shows the potential use of this type of membrane in highly saline media such as seawater.

As for Pb(II), it was observed that by increasing the concentration of NaCl up to 0.3 mol/dm^3^, the transport profiles did not show significant variation for the cases of TEHP and NPOE, while the use of TBEP showed a significant reduction in transport performance. However, a concentration of 0.5 mol/dm^3^ NaCl caused the transport efficiency to decrease in all cases, the transport efficiency for this condition being as TEHP > NPOE > TBEP. This result is associated with the fact that in addition to transporting the metal ion to the receiving phase, NPOE and TBEP plasticizers also allow the transport of chlorides. This fact was qualitatively verified with the addition of five drops of 0.1 mol/dm^3^ AgNO_3_ to 2 cm^3^ of the stripping phase collected at the end of the transport experiments, denoting the presence of the AgCl precipitate.

### 3.4. Effect of the pH in the Feed Phase

In Figure 6, it is observed that there is no significant effect (greater than 5%) in the transport profiles of Cd(II) with the change in pH. However, a significant decrease in the efficiency of Pb(II) transport is observed by increasing the pH of the feed phase to 8.2, obtaining a 64% yield. This effect can be attributed to the low solubility of Pb(II) at that pH value, where the predominant species is Pb(OH)_2_.

### 3.5. Effect of the Matrix Nature

From the transport profiles shown in Figure 7, it is observed that there is no considerable effect on the efficiency of Cd(II) transport at 24 h of pertraction because of the type of matrix with respect to the use of a 0.5 mol/dm^3^ NaCl, pH = 8.2 solution as feed phase, allowing 94% recovery of Cd(II) and 65% of Pb(II). However, the transport kinetics for both metals is considerably affected by this effect. This behavior can be attributed to the saturation of the interface formed between the membrane and the feed phase [56] due to the possible presence of dissolved organic matter (DOM) and other bivalent cations, such as Ca(II), Mg(II), and Sr(II), which are part of seawater. The recovery percentage of the cations were 91% for Cd(II) and 62% for Pb(II) for commercial seawater, showing a small decrease in the extraction percentage compared to synthetic seawater. This similarity may be attributed to the fact that for commercial seawater, a double cold sterilizing microfiltration is reported, which provides a similar composition to synthetic seawater. The results obtained using seawater from Tecolutla, were 86% for Cd(II) and 71% for Pb(II).

As for the case of Pb(II), changes in the transport profiles with the change of matrix were not as significant as for Cd(II). This should be a direct consequence of the complex formed between D2EPHA and the metal ion being much more stable than the complexes formed with the DOM, together with the selectivity of the extractant for Pb(II) over alkali earth metals.

### 3.6. Selective Separation of Cd(II), Pb(II), and Zn(II)

#### 3.6.1. Effect of the Composition of the Feed Solution

When equimolar quantities of the metals were present in F, Cd(II) transport to S1 was efficient and highly selective in all seawater feed phases. The percentage of extraction of the metal at 48 h reached 92% for synthetic seawater, 94% for Panakos seawater, and 90% for seawater from Tecolutla Ver. Mex. Transport of Pb(II) to S2 was highly selective but with a slower kinetics compared to the transport of Cd(II) for all seawater samples. The percentage of extraction of the metal at 48 h reached 96% for synthetic seawater, 93% Panakos seawater, and 87% for seawater from Tecolutla Ver. Mex. The percentages of Zn(II) in the feed phase remained at 90% for synthetic seawater, 93% for Panakos seawater, and 85% for Tecolutla Ver. Mex. No significant difference (greater than 5%) in the transport profiles for the three ions using synthetic seawater and commercial seawater was observed, which can be attributed to the double cold sterilizing microfiltration reported by the seller that allows to separate particles of different nature including macromolecules, suspended solids, colloids, algae, and microorganisms such as bacteria [61], making both samples of seawater practically indistinct. In the case of seawater from Tecolutla beach, there was also no significant decrease (greater than 5%) in the transport of Cd(II). It is known that the predominant soluble species of cadmium on the coasts are chlorides and sulfates or colloidal forms with DOM; however, the affinity for complexation with DOM is much lower than that of copper, lead, and zinc. Cadmium does not easily form complexes with humic and fulvic acids in seawater [62], which explains the low variability between the percentage recovery of Cd(II) using the three seawater samples. On the contrary the decrease in the percentage of transport of Pb(II) using seawater from Tecolutla is directly associated with the formation of soluble complexes of Pb(II) with the possible DOM present in the sample [63].

A simplified transport mechanism for Pb(II) and Cd(II) is depicted in Figure 8. For the sake of clarity, only one species of the metal ions in the PIM phase was considered, although several transport reactions may be present simultaneously, as has been observed in amine transport systems [64]. Clearly as metal ion removal proceeds, the composition in F changes because of the increase in HCl due to the transport of the metals. Accordingly, metal speciation is affected. In Figure 9 the fraction diagrams of the metals as a function of HCl are shown, as calculated using the ChemEQL v3.2 program [65]. The 35% salinity point (0.6 M NaCl) is pointed out to show that the anionic complexes of the different metal ions increase their content as transport continues. However, as previously mentioned (Section 3.2), this is not a necessary condition for transport to occur.

The effect of the initial concentration of ions in F was evaluated, and results are shown in Figure 10 and Figure 11. They show the graphs of the separation factors S(i) of each of the ions in the different phases (F, S1, and S2) for each DOE elemental experiment performed (Table 2). These quantities are defined as
S(Cd) = [Cd]/([Zn] + [Pb])
S(Pb) = [Pb]/([Cd] + [Zn])
S(Zn) = [Zn]/([Pb] + [Cd])

In the DOE, it was observed that by decreasing the initial concentration of the three ions in F, the recovery and transport efficiency increase due to the lower saturation of the active sites of the membranes, this effect being the same for both samples of seawater, Panakos and Tecolutla beach. This observation allows to employ the developed systems in conditions with low initial metal ion concentrations.

Considering competitive experiments, Juang et al. [66] reported the use of Aliquat 336 for the selective separation of Cd(II) and Zn(II) using a liquid membrane in a system with a high concentration of chlorides, concluding that its selective capacity is high if the concentration of zinc is at trace levels, while the selectivity of the extractant decreases with increasing concentration of zinc in the medium. The same effect was reported by Danesi [67] in a supported liquid membrane system. In our case, the values of S(i) and their profiles are highly dependent on the matrix, i.e., Panakos or Tacolutla samples (compare Figure 10 and Figure 11), due to possible differences in DOM content and concomitant metal ions. In addition, metal ion kinetics and the saturation of the active sites of the extractants are influenced by the content of the ions, giving rise to differences in the S(i) profiles.

In the case of the commercial water samples, in experiments A to C, it is observed that the S(Cd) and S(Pb) are favored throughout all times of pertraction, while in experiments D to H, S(Cd) decreases due to the transport of Zn(II) to S1, experiment G being the one with the highest percentage of Zn(II) to that phase. From the transport mechanism (Figure 8) and the fractions diagrams (Figure 9), it is expected that as with the increase in chloride in F, anionic complexes of Zn(II), e.g., $Zn{Cl}_{3}^{-}$, start to compete for the extractant, and once Cd(II) transport is accomplished enough, Zn(II) is transported as well with the consequent decrease in S(Cd) (experiments D–H). Interestingly, migration of Zn(II) to S1 is determined by the composition of F, as in experiments A–C this metal is not transported to S1. This behavior is determined by the magnitude of change in F. Previously, Rojas-Challa et al. [68] have shown that PIM composition determines the magnitude of the flux of HCl from the feed to the strip phase during transport of As(V) with Aliquat 336 as carrier and TEHP as plasticizer, so that PIM composition can be tunned according to the target purpose. Our results now show that the composition of the aqueous phase determines competition of the different metals for the active sites in the PIM, influencing in this form the separation behavior. These results agree with those reported by Hedwig et al. [69] where variations in Sc recovery and selectivity were observed when comparing model solutions and real waste and attributed to the different D2EHPA-to-metal ratios, as in comparison to the model solutions, the industrial acid waste contained more than twice as many elements that were non-equimolar-concentrated. They also agree with results reported by Pyszka and Radzyminska-Lenarcik [37] where the selectivity coefficients in multicomponent mixtures were observed to vary with the quantity of ions in the mixture.

In all experiments, removal of Pb(II) to S2 inhibits its migration to S1. According to the separation profiles obtained for the experiments using seawater from Tecolutla Ver. Mex. (Figure 11), similar conclusions to those previously pointed out can be drawn showing experiments B–D and H as the best separations. Therefore, compositions corresponding to experiments B and C allow S(Cd) and S(Pb)~1000 and 10 < S(Zn) < 1000, depending on the possible amount of DOM and content of concomitant metal ions. As for the values of S(Zn), it is important to remark that lower values in comparison to the other metals are observed, as at the beginning, the three metal ions are present in F, which is not the case for S1 and S2. Additionally, the fact that Zn(II) migrates to S1 once Cd(II) transport has been accomplished allows also the possibility of replacement of S1 with a fresh solution to separate Zn(II) in an additional stage.

Although, in a real scenario, the concentration of the studied heavy metals could vary to a wide extent, the evaluation of seawater speciation at pH 8.1 at the 10^−9^–10^−7^ mol/dm^3^ range of concentrations has shown that 5.8% of Pb(II), 9.8% of Zn(II), and 82.4% of Cd(II) remain complexed with chloride ions [70]. This means that Pb(II) can more easily migrate through the PM system in its free-ion form, and Cd(II) can satisfactorily migrate as well since it is present as a chloride species; only Zn(II) may have reduced transport, requiring probably longer pertraction times.

#### 3.6.2. Stability and Preconcentration Behavior

The constant renewal of F allowed to visualize the stability and the preconcentration capacity of the system (Figure 12). A slight decrease in the removal efficiency was observed after each cycle due to Zn(II) accumulation within the membrane containing Aliquat 336, and in the third cycle a significant loss in selectivity and efficient capacity in S1 was observed, this behavior being the same for both seawater samples. In the case of Pb(II), there is a gradual decrease in transport efficiency after each cycle; however, no loss of selectivity of the membrane containing D2EHPA was observed since no other metal ion was detected in S2. It can also be observed that the system has a high potential for preconcentration, as the driving force is maintained along the cycles, and the concentration of the metals is increased in S1 and S2 after each cycle. Decreasing the volume of the receiving phases can allow a further increase in the concentration of the metals in the recovery phases, a point of great relevance due to the low concentrations of metals in the oceans.

### 3.7. PIMs Characterization

Previous AFM analyses of CTA/NPOE/Aliquat 336 PIMs have shown significant changes in the surface morphology in comparison to PIMs where the plasticizer was not present, showing an increase in surface roughness, which appeared like egg baskets [71]. DSC thermograms showed an increase in the glass transition temperature (T_g_) and the glass transition enthalpy (∆H) values attributed to the incorporation of Aliquat 336 with respect to the polymeric support alone. This suggests that Aliquat 336 may have polar interactions with CTA leading to higher crystalline phases in the PIM. As expected, T_g_ values were also dependent on the plasticizer. The higher ΔH values in NPOE-plasticized membranes indicated a more crystalline phase in the matrices of these membranes [71]. EIS analyses have also revealed that NPOE exhibited a well-defined plasticization effect. The percolation threshold value in the dielectric constant observed with variations in Aliquat 336 concentration suggests that the plasticizer could influence the availability of Aliquat 336 in the membrane. This occurred when the Aliquat 336 formed a mobile fixed site structure or linear reversed micelle arrangement. In the absence of a plasticizer, 50 wt% Aliquat 336 was needed to form reverse micelles in the membrane, whereas in the presence of NPOE, the formation of reversed micelles occurred at 35 wt% and 30 wt% Aliquat, respectively [72].

Similarly, SEM analyses of CTA/D2EHPA/TEHP PIMs have shown that the concentration of phosphorous-containing compounds is constant over the membrane cross-section. They also showed the characteristic fibrillar structure presented by amorphous cellulose derivatives of dense membranes, which was not altered by transport experiments. No asymmetry, pores, or the existence of thin layers supported by macrovoidal structures were detected [55]. AFM images additionally showed a nodular structure and confirmed the presence of cavity channels between the nodules with a nodule diameter of about 45 nm (binodule aggregate about 90 nm) [55].

On the other hand, the ATR-FTIR characterization results of the synthetized PIMs with CTA/NPOE/Aliquat 336 and CTA/D2EHPA/TEHP are presented in Figure 13. The most important signals are listed in Table 4 and Table 5. In general, distinctive bands of the functional groups of the pure membrane components were found in the PIM spectra, and no evidence of the formation of new covalent bonds among them was inferred, leading to the conclusion that the liquid phases (extractant and plasticizer) are encapsulated within the entangled polymeric chain of the base polymer, as expected for this type of membranes [73].

RIMM results shown in Figure 14 and Figure 15 allowed insight into the distribution profiles of the PIM components.

As observed in the CTA/NPOE/Aliquat 336 PIM (Figure 14), NPOE and CTA are almost well distributed along the membrane, confirming the well-defined plasticization effect reported by EIS analyses. Although it was not possible to isolate a pure band for Aliquat 336, by comparing the images for NPOE and NPOE + Aliquat 336, it seems that Aliquat 336 shows defined regions with high extractant content which may imply the formation of reversed micelles, as previously discussed. Although variations in the absorbances along the cross-sections of the images are observed, they are minimal and indicative of very slight changes in composition over the PIM surface.

As for CTA/D2EHPA/TEHP PIMs, the difficulty associated with the isolation of pure bands for the extractant and plasticizer, due to their similitudes in structure, and the strong band overlapping with the base polymer did not allow for measuring individual components, except by CTA. Slightly irregular distributions are observed in comparison to the previous system. This result agrees with that obtained by RIMM analyses showing that the distribution of 2NPOE on the CTA was homogeneous, while those for TBEP and Ionquest*^®^*801, other organophosphate components, were irregular [74]. However, care should be taken in the interpretation of components’ distribution as they are highly dependent on the characteristics of the tertiary mixture [74] and its composition, as reported in PIMs with CTA/TBEP/D2EHPA [75]. Again, the cross-section variations are indicative of very slight changes in composition over the PIM surface.

The results obtained herein support the presence of localized chemical environments due to PIM inhomogeneities that do not necessarily correlate to surface morphology, as described in studies using fluorescence microscopy [76].

### 3.8. Future Perspectives

An implementation using spiral wound modules may be an interesting alternative to explore for the treatment of high volumes of sea water. Additionally, the recovery of more than three metal ions and the final transformation of the recovered metal solutions to obtain high-purity compounds demand further studies. Modeling of the separation factors as a function of metal ion content will facilitate industrial scaling up.

## 4. Conclusions

The synthesis and optimization of PIMs for the individual transport of Cd(II) showed that there was no significant change in the transport profiles and the extraction efficiency when increasing the concentration of NaCl in the feed phase, while the transport profiles for Pb(II) did not show significant variation for the cases of TEHP and NPOE. In contrast, the use of TBEP showed a significant reduction in transport performance, this result being associated with the transport of chlorides from the feed phase. No significant variation in the transport profiles of Cd(II) with the change in pH was observed. However, a significant decrease in the efficiency of Pb(II) transport was observed by increasing the pH of the feed phase to 8.2, this effect being attributed to the low solubility of Pb(II). However, slight changes in the efficiency of Cd(II) transport at 24 h of pertraction were observed because of the matrix nature with respect to the use of a 0.5 mol/dm^3^ NaCl, pH = 8.2 solution as feed phase, and no changes were present in the case of Pb(II); the transport kinetics for both metals were considerably affected. This result is a direct consequence of the possible difference in DOM content and the nature and quantities of the metal ions present in the samples.

The selective separation of Pb(II), Cd(II), and Zn(II) from seawater showed separation factors whose values depended on the composition of the seawater media (metal ion concentrations and matrix composition) in a three-compartment experimental setup with two PIMs, one with Aliquat 336 and another with D2EHPA as carriers. In a typical experiment at 48 h of pertraction with Tecolutla seawater, e.g., with 10^−4^ mol/dm^3^ of each metal, 94% of Zn(II), 90% of Cd(II), and 87% of Pb(II) were present in each phase (F, S1, and S2, respectively) while with 10^−5^ mol/dm^3^ of each metal, 89% of Zn(II), 92% of Cd(II), and 90% of Pb(II) were present in each phase (F, S1, and S2, respectively). Solution compositions corresponding to experiments B and C in Table 2 allow S(Cd) and S(Pb)~1000 and 10 < S(Zn) < 1000, depending on experimental conditions. However, values as high as 10,000 were observed in some experiments, allowing a satisfactory separation of the metal ions. Regardless of the initial composition of the feed solution, satisfactory separations of the metal ions were achieved, although the separation factors’ profiles depended highly on the initial conditions due to the competition for the active sites of the extractant and the differences in transport because of the metal ion content. As excellent transport efficiencies were observed (>99.9%), the system could be used to reduce the content of polluted seawaters to its natural levels. The PIM system was run over three operation cycles of 48 h with satisfactory efficiency. The renewal of the feed phase after each cycle additionally allowed the simultaneous preconcentration of the target metal ions. The conceptualization of the separation system based on the transport mechanisms of the two extractants allowed a chemical interpretation of the observed behavior, in which Pb(II) and Cd(II) migrated to their respective stripping phases, leaving behind Zn(II), which could emigrate later on to one of the stripping phases as the composition of the feed phase changed due to the transport of HCl. Alternatively, Zn(II) could be further recovered by replacement of the stripping phase once Cd(II) transport has finished. It was also observed that best results were obtained when the concentration of the metal ions decreased, as the competition for the active sites did not limit transport, this result being adequate for the treatment of natural sea water samples. The results clearly confirmed the potential of integral and innovative PIM separation schemes for the remediation of marine ecosystems.

## Figures and Tables

**Figure 1 membranes-13-00512-f001:**
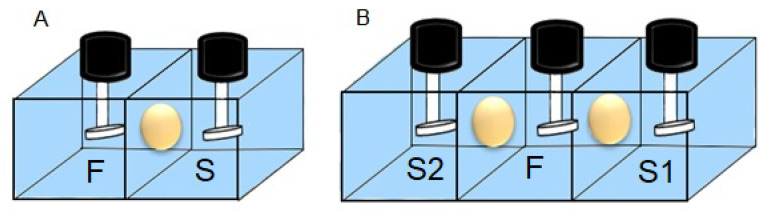
Scheme of home-made cells used for the transport of metal ions through PIMs. (**A**) Two-compartment set-up with one feed (F) and stripping (S) phases; (**B**) Three-compartment set-up with one feed (F) and two stripping (S1, S2) phases.

**Figure 2 membranes-13-00512-f002:**
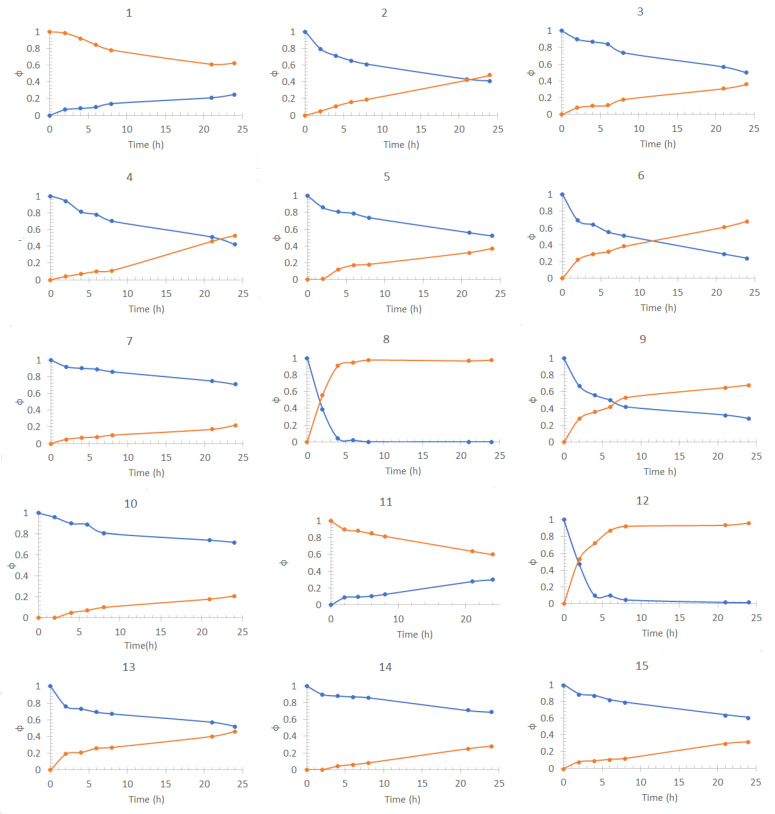
Cd(II) fractions transport profiles in the feed phase (blue dots) and the stripping phase (orange dots) obtained for the Box–Behnken DOE experiments shown in Table 1 (the numbers in the graphs refer to the corresponding experiments). Feed phase: [Metal(II)] = 1 *×* 10*^−^*^4^ mol/dm^3^, pH = 6.5 + 0.1 mol/dm^3^ NaCl. Stripping phase: [HNO_3_] = 0.1 mol/dm^3^.

**Figure 3 membranes-13-00512-f003:**
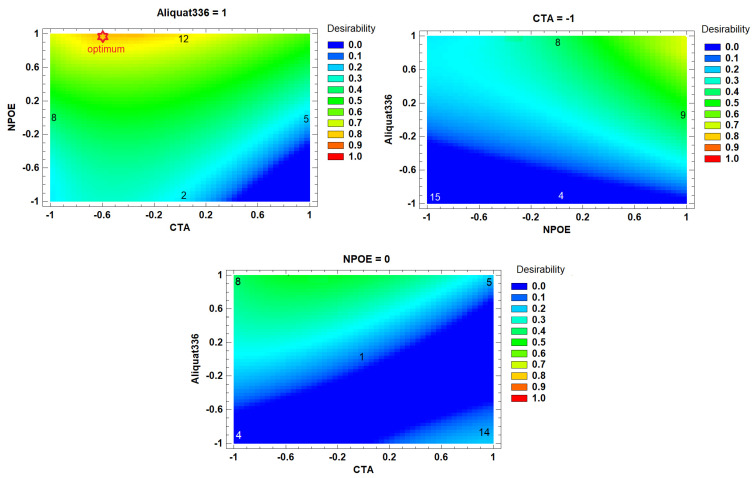
Contour plots of the Cd(II) response surface analysis of the Box–Behnken DOE experiments shown in Table 3. The numbers inside the graphs stand for the elemental experiments reported in the same Table.

**Figure 4 membranes-13-00512-f004:**
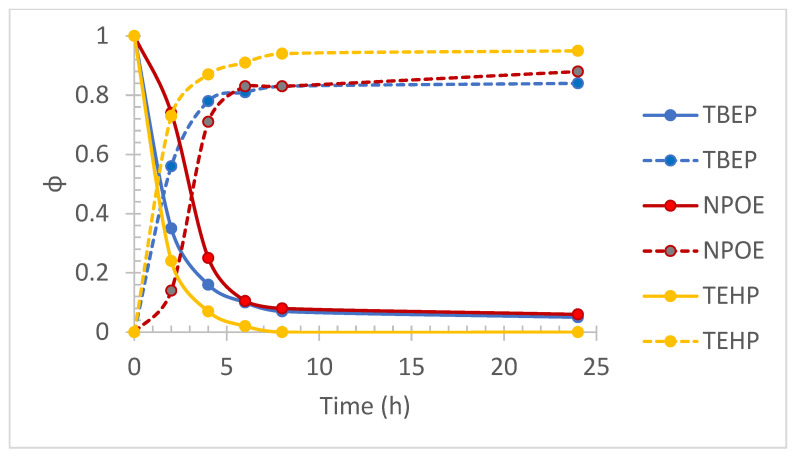
Pb(II) fractions transport profiles in the feed phase (downward curves) and the stripping phase (upward curves) obtained for three different plasticizers (TBEP, NPOE, and TEHP) used for PIM preparation. Feed phase: [Metal(II)] =1 × 10^−4^ mol/dm^3^, pH = 6.5 + 0.1 mol/dm^3^ NaCl. Stripping phase: [HNO_3_] = 0.1 mol/dm^3^.

**Figure 5 membranes-13-00512-f005:**
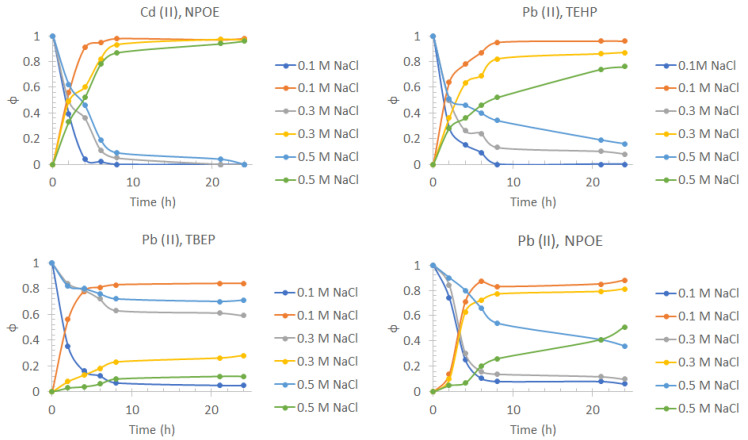
Pb(II) and Cd(II) fractions transport profiles in the feed phase (downward curves) and the stripping phase (upward curves) obtained for the three different plasticizers at different NaCl concentrations in the feed phase. Feed phase: [Metal(II)] = 1 × 10^−4^ mol/dm^3^, pH = 6.5, variable NaCl. Stripping phase: [HNO_3_] = 0.1 mol/dm^3^.

**Figure 6 membranes-13-00512-f006:**
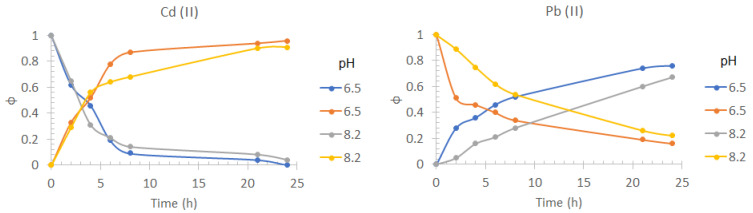
Pb(II) and Cd(II) fractions transport profiles in the feed phase (downward curves) and the stripping phase (upward curves) obtained at two pH values (6.5 and 8.2) in the feed phase. Feed phase: [Metal(II)] = 1 × 10^−4^ mol/dm^3^, variable pH, 0.5 mol/dm^3^ NaCl. Stripping phase: [HNO_3_] = 0.1 mol/dm^3^.

**Figure 7 membranes-13-00512-f007:**
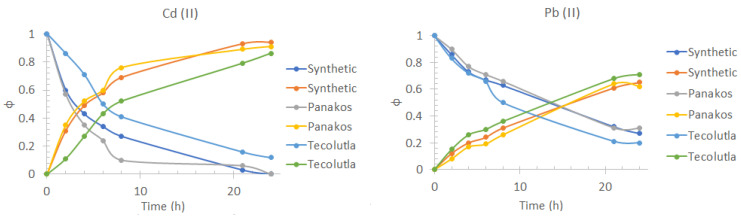
Pb(II) and Cd(II) fractions transport profiles in the feed phase (downward curves) and the stripping phase (upward curves) obtained using synthetic seawater, Panakos commercial seawater, and water collected on Tecolutla beach. Feed phase: [Metal(II)] = 1 *×* 10*^−^*^4^ mol/dm^3^ in different seawater samples. Stripping phase: [HNO_3_] = 0.1 mol/dm^3^.

**Figure 8 membranes-13-00512-f008:**
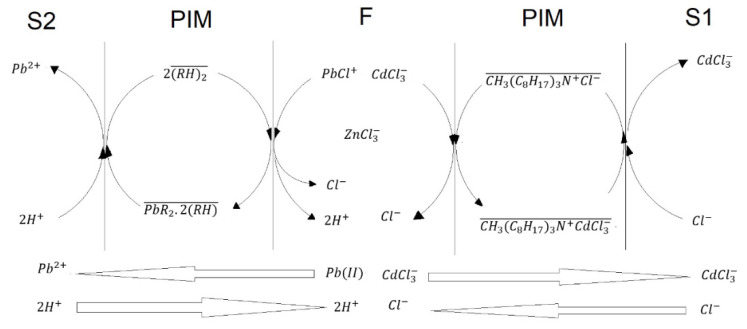
Simplified scheme of the Pb(II) and Cd(II) transport mechanisms in the three-compartment setup.

**Figure 9 membranes-13-00512-f009:**
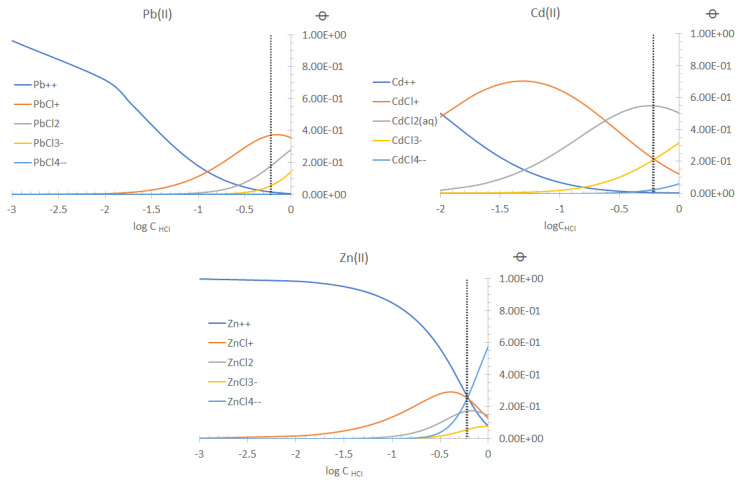
Speciation of the metal ions as a function of the content of HCl medium according to the ChemEQL v3.2 software.

**Figure 10 membranes-13-00512-f010:**
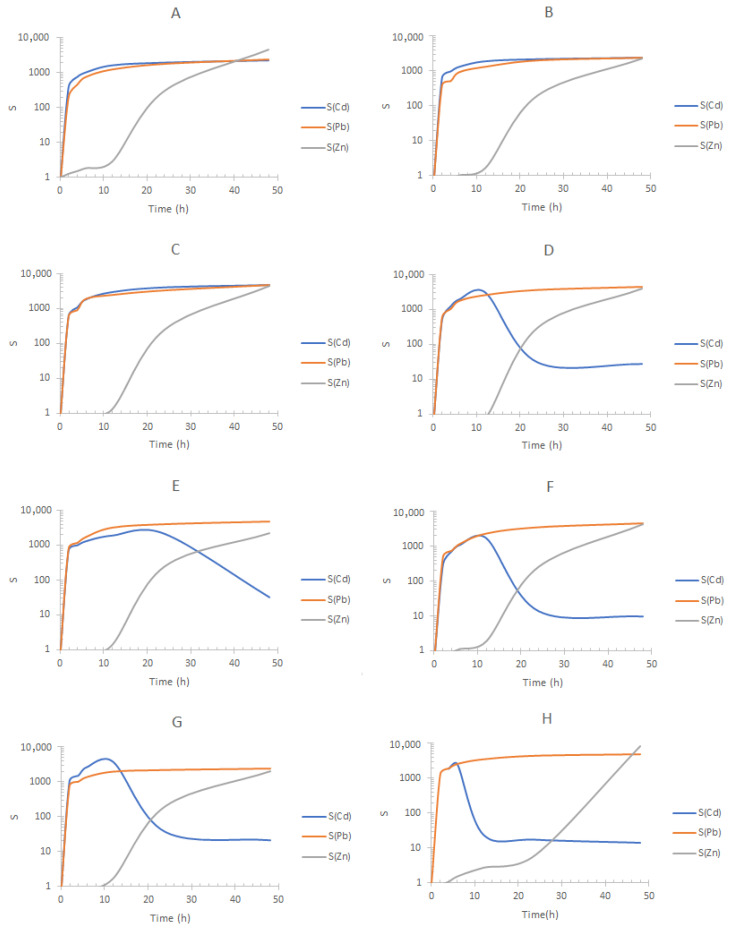
Separation factors (S) of the metal ions for each elemental experiment of the 2^3^ DOE design shown in Table 2 using Panakos sea water and the three-compartment setup. Feed phase for each experiment according to the composition shown in Table 2. Stripping phase S1: 0.1 mol/dm^3^ HCl + 0.1 mol/dm^3^ NaCl; stripping phase S2: 0.1 mol/dm^3^ HNO_3_. The letters of the graphs refer to the experiments reported in Table 2.

**Figure 11 membranes-13-00512-f011:**
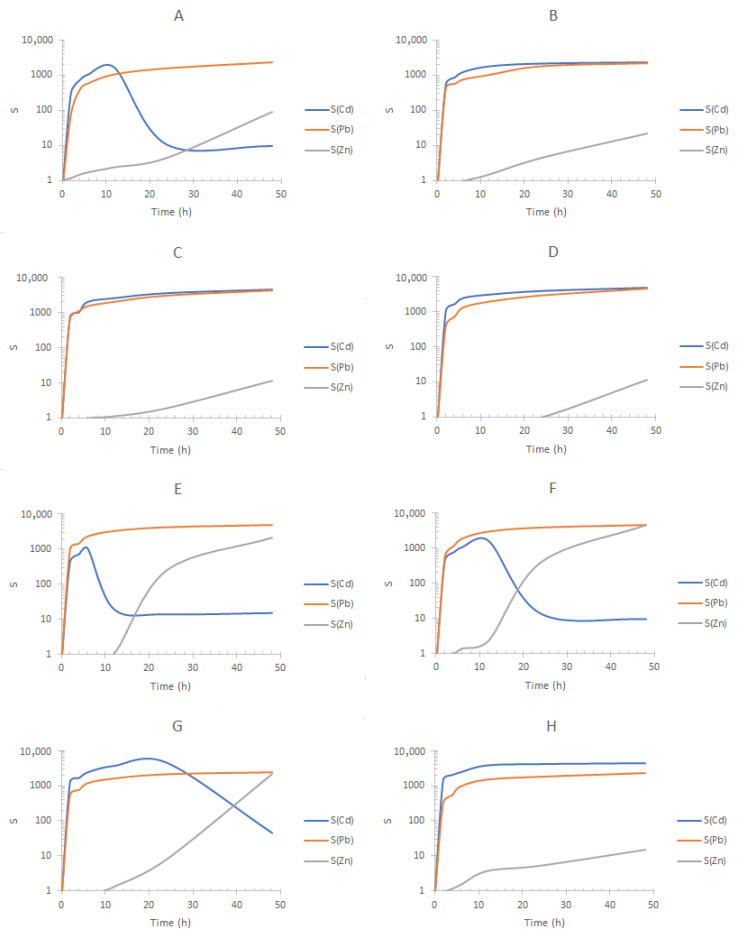
Separation factors (S) of the metal ions for each elemental experiment of the 2^3^ DOE design shown in Table 2 using Tecolutla sea water and the three-compartment setup. Feed phase for each experiment according to the composition show in the Table 2. Stripping phase S1: 0.1 mol/dm^3^ HCl + 0.1 mol/dm^3^ NaCl; stripping phase S2: 0.1 mol/dm^3^ HNO_3_. The letters of the graphs refer to the experiments reported in Table 2.

**Figure 12 membranes-13-00512-f012:**
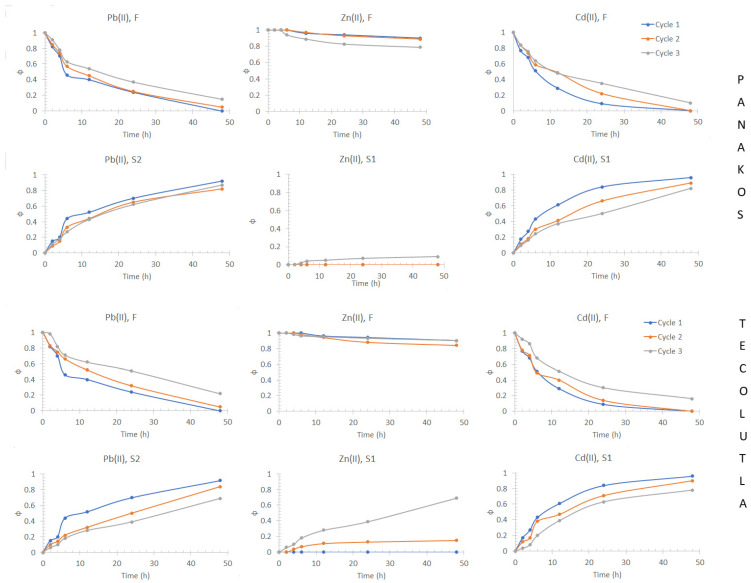
Removal efficiency in the different phases (F, S1, and S2) of the three-compartment setup as a function of the number of cycles of use in which the feed phase was constantly renewed with a fresh solution using two seawater samples. Feed phase: 1 × 10^−4^ mol/dm^3^ metal(II); stripping phase S1: 0.1 mol/dm^3^ HCl + 0.1 mol/dm^3^ NaCl; stripping phase S2: 0.1 mol/dm^3^ HNO_3_.

**Figure 13 membranes-13-00512-f013:**
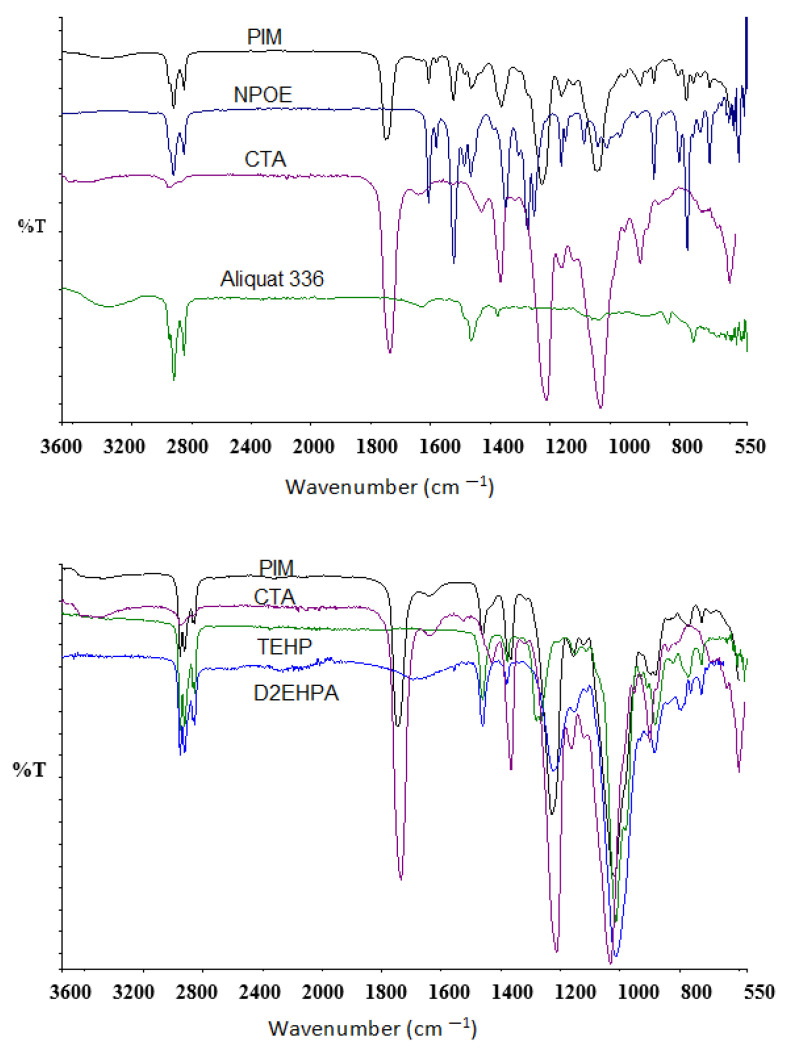
ATR-FTIR spectra of the CTA/NPOE/Aliquat 336 (**upper**) and CTA/D2EHPA/TEHP (**lower**) PIMs and their pure components.

**Figure 14 membranes-13-00512-f014:**
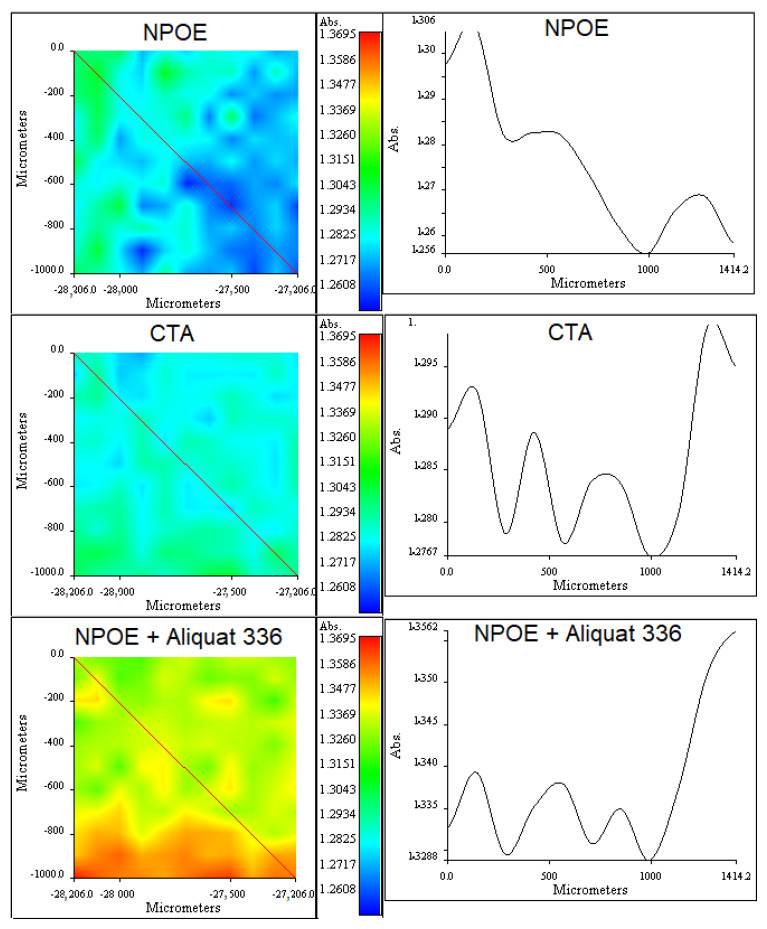
(**Right column**): RIMM maps obtained for NPOE (1520 cm^−1^), CTA (1750 cm^−1^), and NPOE + Aliquat 336 (2923 cm^−1^). (**Left column**): cross-section showing the variations of the measured absorbances along the lines indicated in the maps.

**Figure 15 membranes-13-00512-f015:**
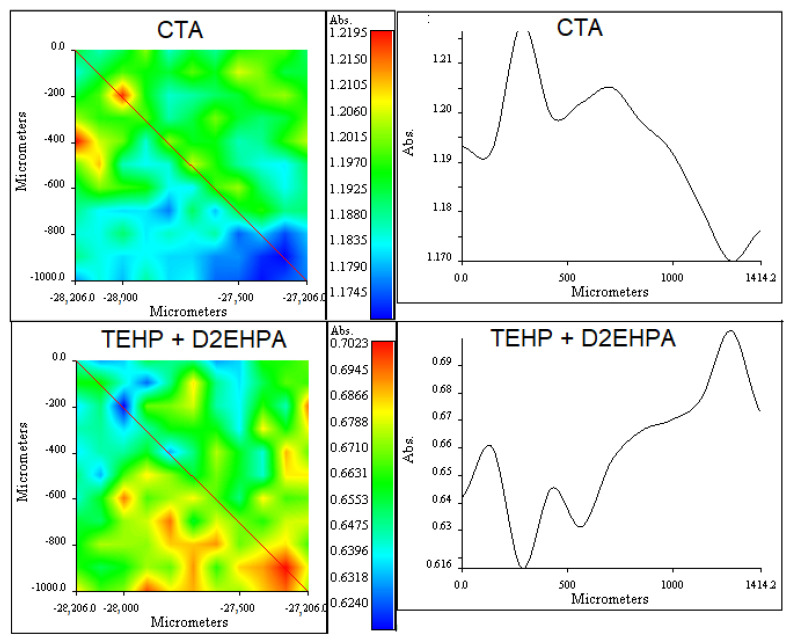
(**Right column**): RIMM maps obtained for CTA (1750 cm^−1^) and TEHP + D2EHPA (727 cm^−1^). (**Left column**): cross-section showing the variations of the measured absorbances along the lines indicated in the maps.

**Table 1 membranes-13-00512-t001:** Code and real values for the Box–Behnken DOE design.

Coded Value	CTA (g)	NPOE (g)	Aliquat 336 (g)
1	0.15	0.0475	0.12
0	0.10	0.0325	0.08
−1	0.05	0.0175	0.04

**Table 2 membranes-13-00512-t002:** Experimental design array in coded values used for selectivity experiments. Real values are in parentheses.

Experiment	Zn(II) (mol/dm^3^)	Pb(II) (mol/dm^3^)	Cd(II) (mol/dm^3^)
A	1 (1 × 10^−4^)	−1 (5 × 10^−5^)	−1 (5 × 10^−5^)
B	−1 (5 × 10^−5^)	−1 (5 × 10^−5^)	−1 (5 × 10^−5^)
C	1 (1 × 10^−4^)	1 (1 × 10^−4^)	1 (1 × 10^−4^)
D	−1 (5 × 10^−5^)	1 (1 × 10^−4^)	1 (1 × 10^−4^)
E	−1 (5 × 10^−5^)	1 (1 × 10^−4^)	−1 (5 × 10^−5^)
F	1 (1 × 10^−4^)	1 (1 × 10^−4^)	−1 (5 × 10^−5^)
G	−1 (5 × 10^−5^)	−1 (5 × 10^−5^)	1 (1 × 10^−4^)
H	1 (1 × 10^−4^)	−1 (5 × 10^−5^)	1 (1 × 10^−4^)

**Table 3 membranes-13-00512-t003:** Transport results for the Cd(II) DOE experiments (24 h of pertraction time).

Experiment	CTA	NPOE	Aliquat 336	Φ*_F_*	Φ*_S_*
1	0	0	0	0.25	0.63
2	0	−1	1	0.41	0.48
3	0	0	0	0.5	0.36
4	−1	0	−1	0.42	0.53
5	1	0	1	0.52	0.37
6	1	1	0	0.24	0.68
7	0	−1	−1	0.71	0.22
8	−1	0	1	Nd *	0.98
9	−1	1	0	0.28	0.68
10	1	−1	0	0.72	0.21
11	0	0	0	0.3	0.6
12	0	1	1	0.014	0.96
13	0	1	−1	0.52	0.46
14	1	0	−1	0.69	0.28
15	−1	−1	0	0.61	0.32

* Not detectable.

**Table 4 membranes-13-00512-t004:** Characteristic vibrations for the functional groups in the CTA/NPOE/Aliquat 336 PIM.

Group	Wavenumber (cm^−1^)	PIM	CTA	Aliquat 336	NPOE
C=O	1750	X	X		
C-H	2951	X	X		X
OH	3600		X		
CH_3_	1368	X	X	X	
COO	1649		X		
C–NO_2_	1520	X			X
COO	1434		X		
-C-O-C-	1217	X	X		
-C-O-C-	1041	X	X		
C-O	1279		X		X
C-O	1165				X
N-O	1525	X			X
N-O	744				X
-C=C-	1468				X
-C=C-	1608				X
=C-H	2923				X
N-C	3373	X		X	
CH_2_	1466			X	
N-C	2923	X		X	

**Table 5 membranes-13-00512-t005:** Characteristic vibrations for the functional groups in the CTA/D2EHPA/TEHP PIM.

Group	Wavenumber (cm^−1^)	PIM	D2EHPA	TEHP	CTA
CH	2959, 881	X	X	X	X
CH_2_	1465	X	X	X	X
CH_3_	1379	X	X	X	X
P=O	1282		X	X	
P-O-C	1022	X	X	X	
P-O-H	2282		X		
Hydrogen bond	1680		X		
P=O	1221	X	X	X	
C=O	1750	X			X
OH	3600				X
COO	1649				X
COO	1520				X
COO	1434				X
-C-O-C-	1217				X
-C-O-C-	1041				X
CH_2_	727	X	X	X	

## Data Availability

The data that support the findings of this study are available upon reasonable request.

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
