# Peer review of "On the Use of Polymer Inclusion Membranes for the Selective Separation of Pb(II), Cd(II), and Zn(II) from Seawater"

_membranes, 2023, doi:10.3390/membranes13050512_

Round 1

Reviewer 1 Report

 Attached file

-

Author Response

Reviewer 1

The authors have presented a study based on the selective separation of Pb(II), Cd(II), and Zn(II) from seawater by PIMs.In addition to some specific problems, I understand that the work has a major problem that affects its main objective. Although the work is interesting, in fact the selective separation from seawater of the chemical species mentioned in the title has not been carried out.

The authors have used up to three seawater samples (one artificial, one commercial realsample, and one real seawater sampled form the Tecolutla beach). As the authors mention in the manuscript Cd, and Pb appears at the ppt levels in real seawater (up toabout 10-8-10-9 M), while Zn may appears at the low ppb level (up to about 10-8-10-9 M).Nevertheless, the authors added metals to seawater up to 10-4 M (up to 100,000 times higher) and then, the chemical behavior of the samples was completely different than in seawater.

Response: The objective of the work was not to transport the metals at the natural levels that they are present in sea water, but from simulated polluted conditions with a serious potential ecological risk index. This has now been clarified in the last paragraph of the introduction section (lines 139-154). It is the intention of the work to reduce the metal concentrations of this polluted samples up to its natural levels.

As the authors explained, chemical speciation is essential to understand the processes taking place in the membrane systems studied, especially with regard to the formation of chlorocomplexes. However, the chemical speciation of Cd and Pb in seawater (which is in fact not limited to CdCl3- and Pb2+) is highly influenced by the metal concentration, and will be different at a concentration of 10-4 M and at 10-8 M. We could say the same for Cl- and when authors did some experiments with 0.1M Cl- and others with 0,5 M Cl-, they have in fact different species. As may be seen in the figure (taken from the book Environmental Chemistry from vanLoon and Duffy), at 0.1M (at environmental concentrations) more than 60% is a cationic species (CdCl+), followed by the neutral CdCl2, and CdCl3- is less than 10%. At 0.5M, CdCl3- and CdCl2 are about 30% each one. The behavior of these species may be different in the membrane systems.

Something similar occurs with the interferences produced by other metals, especially Ca and Mg, which are the main problem in seawater and will interfere in a very different manner if target metals are at a concentration of 10-4 or 10-8 M. In fact, in my opinion, the developed systems would not behave in the same way with seawater containing metals at relevant environmental concentrations.

Just as an example, I am not sure about the results explained by the authors as a function of DOM (which was not measured), and they could be related with different concentrations of the metal chemical species and/or Ca and Mg in waters.

My recommendation is to focus the manuscript on a membrane system to separate highconcentrations of Cd and Pb from saline matrices and, as a previous step, to calculatechemical speciation in each of the real conditions used by the authors. Then, to revise discussion accordingly

Response: The chemical speciation was evaluated at the experimental conditions employed, i.e., high metal concentrations. The speciation diagrams were evaluated considering the simultaneous changes in proton and chloride ions observed during experimentation, and not only at fixed pH value and variable concentrations of chloride as in the Fig. 10.2. We agree that speciation changes with concentrations, and for this reason the discussion is center at the employed experimental conditions. Although changes in speciation may change the transport mechanism, as explained in the text, the proposed one is just a simplified version, as other reactions may also be present. This is now included in lines 452-460. However, as changes in speciation may impact the separation behavior, a comment about this situation has been added in lines 538-544.

Additionally, some minor comments:

- More quantitative information about the results obtained should be introduced in the abstract

Response: Done.

- Reference 4 is not the best one for the definition of heavy metals.

Response: the reference was changed.

- References 3 and 7 are the same.

Response: This error was corrected.

- References for EPA’s data are missing (lines 52 and 58).

Response: Done (lines 53 and 59).

- Reference for Zn concentrations is required (line 70).

Response: Done (line 67).

- In lines 97, the authors use terms such as removal or remediation. Is a realistic approach with a PIM?

Response: This word has been removed considering that these are only a laboratory experiments.

- In lines 125-128, please specify reference for commercial seawater, correct 0.22 microns, and specify how the Tecolutla sample was taken and preserved.

Response: Done. The procedures and a reference were added (lines 178-182).

- In section 2.2, please explain if the thickness and porosity of the membrane were measured and how they affect to the reproducibility.

Response: The characterization of the PIM was outside the scope of the paper. However, the PIM with D2EHPA has already been characterized. This in now indicated in lines 225-228.

- In figure 1, the position of each solution should be indicated.

Response: Done.

- Section 2.4 is not clear to me. Why only one membrane was optimized?

Response: Because the other membrane was used as previously reported without any change. The reported composition gave excellent results without further optimization.

- Please correct some errors along the text, such as DE2HPA instead of D2EHPA, DDE instead of DOE, % instead of ‰, etc.

Response: Done.

- In figure 2, the colors of some lines are exchanged.

Response: The graphic was redrawn.

- If an optimization with DOE has been done, why is the highest experiment value chosen instead of making a selection of the optimum, for example with a response surface?

Response: For the sake of simplicity the RSM analysis was not previously included as it provides the same results obtained with the maxima. However, now this analysis was included (lines 293-331 and Figure 3).

- As mentioned before, in 3.2, a partial selection of species has been done.

Response: Yes, at this is only a simplified scheme.

- Why the optimization was done only for Cd and Pb (not for Zn)?

Response: To transport Zn once Cd transport has been finished using the same PIM.

- Errors bars should be included in all the figures. Some of the results discussed could be affected by reproducibility, for example, in figure 5.

Response: We agree, but to make the Figures no more complex a statement concerning the error has been added (line 212).

- If the authors use DOM to explain some results, data of DOM concentrations should be included.

Response: as DOM was not measured, we have indicated “possible differences in DOM content”.

Reviewer 2 Report

The topic of submitted manuscript is very relevantand could be interested to the auditory of the Membrsnes journal. here are some points which must be edited or clarified by providing additional information or a comment. This manuscript is recommended to be published only after including and addressing the below listed comments with major revision 

1. The authors should clearly explain the innovation and importance of their work on the introduction of the manuscript. They should justify the value of the work and compare their work with previously similar published papers. Please add more recent references related to proposed manuscript.

2. The main lack of this study is the absence of any experimental data described morphology  and structure/composition of prepared PIMs. Authors have to provide missing detailed information in revised manuscript. I suggest to add some SEM data, as well as FTIR, EDX et al.

3. The quality of figures #3,4,6-11 should be significantly improved. In presence

4. Authors have to add comparative data on prepared PIMs  and other types of membranes fom related publications.

5. What about statistical analysis in all experiments on membranes examination?

6. The conclusion section should be elaborated and improved. The author should bring specific conclusions in accordance with obtained results.

Our decision on this manuscript – Major revision. After making substantial changes in article it could be recommended for publication

Author Response

Reviewer 2

The topic of submitted manuscript is very relevant and could be interested to the auditory of the Membranes journal. Here are some points which must be edited or clarified by providing additional information or a comment. This manuscript is recommended to be published only after including and addressing the below listed comments with major revision.

  1. The authors should clearly explain the innovation and importance of their work on the introduction of the manuscript. They should justify the value of the work and compare their work with previously similar published papers. Please add more recent references related to proposed manuscript.

Response: The introduction section was completely rewritten to address this observation. In the last paragraph (lines 139-154) the innovation was clearly indicated.

  1. The main lack of this study is the absence of any experimental data described morphology and structure/composition of prepared PIMs. Authors have to provide missing detailed information in revised manuscript. I suggest adding some SEM data, as well as FTIR, EDX et al.

Response: The characterization of the PIM was outside the scope of the paper. However, the PIM with D2EHPA has already been characterized. This in now indicated in lines 225-228.

  1. The quality of figures #3,4,6-11 should be significantly improved.

Response: Done.

  1. Authors have to add comparative data on prepared PIMs and other types of membranes fom related publications.

Response: Comparisons to other PIM systems is now indicated in lines 514-527.

  1. What about statistical analysis in all experiments on membranes examination?

Response: A statement concerning this observation has been added (line 212).

  1. The conclusion section should be elaborated and improved. The author should bring specific conclusions in accordance with obtained results.

Response: Done.

 Our decision on this manuscript – Major revision. After making substantial changes in article it could be recommended for publication

Reviewer 3 Report

Detailed comments:

1.      The English of the text should be checked

2.      The novelty of the manuscript is complete missing

3.      In the Introduction part must be included more information about techniques for the quantification, extraction or remediation of aqueous media in the presence of heavy metals  (advantages and disadvantages, comparison with other methods), about polymeric inclusion membranes (e.g., electrodialysis, reverse osmosis, ultrafiltration, nanofiltration). The following recommended references could be included in the Introduction part to improve the quality of manuscript, because they provide relevant information:

ü  Membrane Technologies in Wastewater Treatment: A Review. Membranes 2020, 10, 89

ü  Advanced Hybrid Membranes for Efficient Nickel Retention from Simulated Wastewater, Polymer International, 2021, 70(6), p. 866-876

ü  Effective removal of heavy metal ions Cd2+, Zn2+, Pb2+, Cu2+ from aqueous solution by polymer-modified magnetic nanoparticles. J. Hazard. Mater. 2012, 211–212, 366–372

ü  Adsorbents/ion exchangers-PVA blend membranes: Preparation, characterization and performance for the removal of Zn2+ by electrodialysis, Applied Surface Science 329, 2015, p. 65-75

4.      Authors write: “The presence of heavy metals in the environment…”; “Heavy metals accumulate in animals and plants…” – example of heavy metals must be included/mentioned

5.      At Figures 2, 3, 4, 5, 6, 9, 10 and 11 for Time indicate unit of measure. Also, at scale for division the line, use the option “Tick marks”

6.      For all parameters used in the equations indicate what represent

7.      For all abbreviations or notation indicate the complete name

8.      For unit of measure used the S.I., please check in all manuscript.

9.      For all operational conditions must be indicated, amount, concentration, time, speed rotation

10.  For all reagents or chemicals used must be indicated manufacturer, purity, concentration, amount

11.  All equipment and tools used in this study should be described in detail or further information should be provided (manufacturer, type, operational conditions, etc.)

12.  More Conclusions with the best result obtained

13.  Comparison between the obtained results and measured in this study with other reported studies should be done and included for more clarity (indicate values not just number of reference).

14.  The potential application of the prepared and used materials must be indicated in the Conclusion part

15.   “Challenges or Future Prospective” is missing in the manuscript

Detailed comments:

1.      The English of the text should be checked

2.      The novelty of the manuscript is complete missing

3.      In the Introduction part must be included more information about techniques for the quantification, extraction or remediation of aqueous media in the presence of heavy metals  (advantages and disadvantages, comparison with other methods), about polymeric inclusion membranes (e.g., electrodialysis, reverse osmosis, ultrafiltration, nanofiltration). The following recommended references could be included in the Introduction part to improve the quality of manuscript, because they provide relevant information:

ü  Membrane Technologies in Wastewater Treatment: A Review. Membranes 2020, 10, 89

ü  Advanced Hybrid Membranes for Efficient Nickel Retention from Simulated Wastewater, Polymer International, 2021, 70(6), p. 866-876

ü  Effective removal of heavy metal ions Cd2+, Zn2+, Pb2+, Cu2+ from aqueous solution by polymer-modified magnetic nanoparticles. J. Hazard. Mater. 2012, 211–212, 366–372

ü  Adsorbents/ion exchangers-PVA blend membranes: Preparation, characterization and performance for the removal of Zn2+ by electrodialysis, Applied Surface Science 329, 2015, p. 65-75

4.      Authors write: “The presence of heavy metals in the environment…”; “Heavy metals accumulate in animals and plants…” – example of heavy metals must be included/mentioned

5.      At Figures 2, 3, 4, 5, 6, 9, 10 and 11 for Time indicate unit of measure. Also, at scale for division the line, use the option “Tick marks”

6.      For all parameters used in the equations indicate what represent

7.      For all abbreviations or notation indicate the complete name

8.      For unit of measure used the S.I., please check in all manuscript.

9.      For all operational conditions must be indicated, amount, concentration, time, speed rotation

10.  For all reagents or chemicals used must be indicated manufacturer, purity, concentration, amount

11.  All equipment and tools used in this study should be described in detail or further information should be provided (manufacturer, type, operational conditions, etc.)

12.  More Conclusions with the best result obtained

13.  Comparison between the obtained results and measured in this study with other reported studies should be done and included for more clarity (indicate values not just number of reference).

14.  The potential application of the prepared and used materials must be indicated in the Conclusion part

15.   “Challenges or Future Prospective” is missing in the manuscript

Author Response

Reviewer 3

Detailed comments:

  1. The English of the text should be checked

Response: Done.

  1. The novelty of the manuscript is complete missing

Response: The introduction section was completely rewritten to address this observation. In the last paragraph (lines 139-154) the innovation was clearly indicated.

  1. In the Introduction part must be included more information about techniques for the quantification, extraction or remediation of aqueous media in the presence of heavy metals (advantages and disadvantages, comparison with other methods), about polymeric inclusion membranes (e.g., electrodialysis, reverse osmosis, ultrafiltration, nanofiltration). The following recommended references could be included in the Introduction part to improve the quality of manuscript, because they provide relevant information:

ü  Membrane Technologies in Wastewater Treatment: A Review. Membranes 2020, 10, 89

ü  Advanced Hybrid Membranes for Efficient Nickel Retention from Simulated Wastewater, Polymer International, 2021, 70(6), p. 866-876

ü  Effective removal of heavy metal ions Cd2+, Zn2+, Pb2+, Cu2+ from aqueous solution by polymer-modified magnetic nanoparticles. J. Hazard. Mater. 2012, 211–212, 366–372

ü  Adsorbents/ion exchangers-PVA blend membranes: Preparation, characterization and performance for the removal of Zn2+ by electrodialysis, Applied Surface Science 329, 2015, p. 65-75

 Response: Although a complete comparison between membranes and other separation methods is outside the scope of the manuscript, new information concerning this point has been added in lines 68-100.

  1. Authors write: “The presence of heavy metals in the environment…”; “Heavy metals accumulate in animals and plants…” – example of heavy metals must be included/mentioned

Response: This phrase is not longer present in the revised version.

  1. At Figures 2, 3, 4, 5, 6, 9, 10 and 11 for Time indicate unit of measure. Also, at scale for division the line, use the option “Tick marks”

Response: Done.

  1. For all parameters used in the equations indicate what represent

Response: Done.

  1. For all abbreviations or notation indicate the complete name

Response: Done.

  1. For unit of measure used the S.I., please check in all manuscript.

Response: Done.

  1. For all operational conditions must be indicated, amount, concentration, time, speed rotation

Response: We consider that specific information concerning each of the experiments is described in sections 2.5-2.8. General information concerning transport setup is described in section 2.3.

  1. For all reagents or chemicals used must be indicated manufacturer, purity, concentration, amount

Response: Done.

  1. All equipment and tools used in this study should be described in detail or further information should be provided (manufacturer, type, operational conditions, etc.)

Response: Done.

  1. More Conclusions with the best result obtained

Response: Done.

  1. Comparison between the obtained results and measured in this study with other reported studies should be done and included for more clarity (indicate values not just number of reference).

Response: Comparisons to other PIM systems is now indicated in lines 514-527. As there are not similar works with PIM for seawater, the discussion was maintained only at a qualitative level.

  1. The potential application of the prepared and used materials must be indicated in the Conclusion part

Response: Comparisons to other PIM systems is now indicated in lines 514-527.

  1. “Challenges or Future Prospective” is missing in the manuscript

Response: A new section “3.7 Future Perspectives” was added.

Round 2

Reviewer 1 Report

In the revised version of the manuscript, the authors have responded to some of the questions raised, but other important questions remained unresolved. In this sense, the manuscript still focus on the selective separation of Pb, Cd and Zn from seawater, something that, in my opinion, has not been done in this work. Some other important unresolved issues are, for example, the way to choose the optimal values ​​(the experiment with the best response is still simply selected) or the role of the different chemical species. Additionally, there are still some pending corrections that have not been made, such as the consistency of the curves in figure 2, the correct expression of salinity, etc. More examples could be mentioned.

.

Author Response

In the revised version of the manuscript, the authors have responded to some of the questions raised, but other important questions remained unresolved. In this sense, the manuscript still focus on the selective separation of Pb, Cd and Zn from seawater, something that, in my opinion, has not been done in this work. Some other important unresolved issues are, for example, the way to choose the optimal values (the experiment with the best response is still simply selected) or the role of the different chemical species. Additionally, there are still some pending corrections that have not been made, such as the consistency of the curves in figure 2, the correct expression of salinity, etc. More examples could be mentioned.

Answer:

The phrase “ In a typical experiment at 48 h of pertraction with Tecolutla seawater, e.g., with 10-4 mol/dm3 of each metal, 94% of Zn(II), 90% of Cd(II), and 87% of Pb(II) were present in each phase (F, S1, and S2, respectively) while with 10-5 mol/dm3 of each metal, 89% of Zn(II), 92% of Cd(II), and 90% of Pb(II) were present in each phase (F, S1, and S2, respectively)” was added in lines 712-717 to exemplify the selective behavior of the system.

The way in which optimal values were selected is indicated in lines 331-336: “In Fig. 3 contour plots of the desirability function are shown. Intermediate to high NPOE content, with high Aliquat 336 and low CTA content makes the best PIMs. According to the response surface, an optimal desirability value of 0.81 may be achieved at -0.60 CTA, 1.0 NPOE, 1.0 Aliquat 336 PIM coded composition. However, this PIM was fragile and difficult to manipulate. Membrane 8 of Table 3 (-1.0 CTA, 0.0 NPOE, 1.0 Aliquat 336) reached a desirability of 0.77, representing a non-significant difference with respect to the optimum value”.

The role of chemical species has been discussed in lines 373-389: “However, to be precise, historically the term anion exchange originates from salt metathesis reactions, in which the anion initially present in the organic phase is exchanged by another anion initially present in the aqueous phase. The extraction of metals by basic extractants is usually assumed to be facilitated by the formation of the anionic MXy^(n-y) complex (M:metal, X: complexing agent, e.g., chloride) in the aqueous phase. However, experimental evidence has shown that the presence of the negatively charged anions in the aqueous phase is not mandatory, as the metal can be extracted to the organic phase in spite that such species are not present in the aqueous phase. A new extraction model has then been provided. This model relies on the hypothesis that the metal species least stabilized in the aqueous phase by hydration (i.e., the metal species with the lowest charge density) is extracted more efficiently than the more water-stabilized species (i.e., species with higher charge densities). Once it is transferred to the organic phase, the extracted species can undergo further Lewis acid−base adduct formation reactions with the chloride anions available in the organic phase to form negatively charged chloro-complexes in that phase, i.e., the anionic compounds are directly formed in the organic phase without requiring to be present initially in the aqueous phase [59]” and in lines 574-580: “Although in a real scenario, the concentration of the studied heavy metals could vary in a wide extent, the evaluation of seawater speciation at pH 8.1 at the 10-9 – 10-7 mol/dm3 range of concentrations have shown that 5.8 % of Pb(II), 9.8% of Zn(II), and 82.4% of Cd(II) remains complexed with chloride ions [70]. This means that Pb(II) can more easily migrate through the PM system in its free-ion form, that Cd(II) can satisfactorily migrate as well as it is present as a chloride species, and only Zn(II) may have a reduced transport, requiring probably longer pertraction times”

The consistency of the curves in figure 2 was revised.

Sea water salinity is expressed as a ratio of salt (in grams) to liter of water. In sea water there is typically close to 35 grams of dissolved salts in each liter. It is written as 35%.  The normal range of ocean salinity ranges between 33-37 grams per liter (33 – 37%). Sorry, but we don´t understand the comment.

More examples concerning possible alternatives for the removal of metals are mentioned in lines 68-71: “Although different technologies for removing heavy metal ions from waters are currently available, e.g., chemical precipitation, ion-exchange, adsorption, coagulation–flocculation, flotation, photocatalysis, electrochemical methods [14, 15] and polymer-modified magnetic nanoparticles [16],…”

Reviewer 2 Report

The authors provided significant improvements in the revised manuscript. Nevertheless, the main query was not answered. It is a very important point related to the characterization of used membranes due it was not commercial samples but laboratory prepared. Moreover, these samples have to be characterized prior to testing. This critical point must be clarified prior manuscript will be recommended for publication.

Author Response

The authors provided significant improvements in the revised manuscript. Nevertheless, the main query was not answered. It is a very important point related to the characterization of used membranes due it was not commercial samples, but laboratory prepared. Moreover, these samples have to be characterized prior to testing. This critical point must be clarified prior manuscript will be recommended for publication.

Answer:

The new section “3.7 PIM characterization” was added and a discussion concerning the actual literature in this aspect was included.

Reviewer 3 Report

Agree

Author Response

Thank you for your helpful comments.

Round 3

Reviewer 2 Report

Finally authors provides detailed description of studied PIMs in revised manuscript thus it can be recommended for publication.